# Potential Benefits of Selenium Supplementation in Reducing Insulin Resistance in Patients with Cardiometabolic Diseases: A Systematic Review and Meta-Analysis

**DOI:** 10.3390/nu14224933

**Published:** 2022-11-21

**Authors:** Jiahui Ouyang, Yajie Cai, Yewen Song, Zhuye Gao, Ruina Bai, Anlu Wang

**Affiliations:** 1Xiyuan Hospital, China Academy of Chinese Medical Sciences, Beijing 100091, China; 2National Clinical Research Center for Chinese Medicine Cardiology, Xiyuan Hospital, China Academy of Chinese Medical Sciences, Beijing 100091, China

**Keywords:** selenium, cardiometabolic disease, insulin resistance, diabetes mellitus, cardiovascular disease, systematic review, meta-analysis

## Abstract

Background: Selenium is a trace element that has been reported to be effective in regulating glucose and lipid metabolism. However, there is conflicting evidence from different clinical trials of selenium supplementation in treating cardiometabolic diseases (CMDs). Objective: This meta-analysis aimed to identify the effects of selenium supplementation on insulin resistance, glucose homeostasis, and lipid profiles in patients with CMDs. Methods: Randomized controlled trials (RCTs) of selenium supplementation for treating CMDs were screened in five electronic databases. Insulin levels, homeostatic model assessment of insulin resistance (HOMA-IR), fasting plasma glucose (FPG), and glycosylated hemoglobin A1C (HbA1c) were defined as the primary outcome markers, and lipid profiles were considered the secondary outcome markers. Results: Ten studies involving 526 participants were included in the meta-analysis. The results suggested that selenium supplementation significantly reduced serum insulin levels (standardized men difference [SMD]: −0.53; 95% confidence interval [CI] [−0.84, −0.21], *p* = 0.001, *I*^2^ = 68%) and HOMA-IR (SMD: −0.50, 95% CI [−0.86, −0.14], *p* = 0.006, *I*^2^ = 75%) and increased high-density lipoprotein cholesterol (HDL-C) levels (SMD: 0.97; 95% CI [0.26, 1.68], *p* = 0.007, *I*^2^ = 92%), but had no significant effect on FPG, total cholesterol (TC), triglycerides (TG), low-density lipoprotein cholesterol (LDL-C), and very low-density lipoprotein cholesterol (VLDL-C). Conclusion: Current evidence supports the beneficial effects of selenium supplementation on reducing insulin levels, HOMA-IR, and increasing HDL-C levels. Selenium supplementation may be an effective strategy for reducing insulin resistance in patients with CMDs. However, more high-quality clinical studies are needed to improve the certainty of our estimates.

## 1. Introduction

Cardiometabolic diseases (CMDs) begin with clinically high-risk states ranging from insulin resistance (IR) to prediabetes states (e.g., obesity) and metabolic syndrome (MS), which then progress to type 2 diabetes mellitus (T2DM) and cardiovascular disease (CVD) [1]. Most individuals diagnosed with CMDs also experience additional cardiometabolic risks, including obesity, dyslipidemia, hyperinsulinemia, and thrombosis [2]. Unhealthy diets and accelerated population aging have resulted in an annual increase in the prevalence and mortality of CMDs, which places a significant economic burden on the healthcare systems of various countries [3,4]. Studies have shown that these metabolic disorders may be influenced by nutrition. Individual nutrient and dietary supplements composed of sufficient nutrients may be associated with reduced cardiometabolic risks, suggesting that nutritional intervention measures are effective in managing these diseases [5,6,7,8].

Selenium is a micronutrient that is vital for human health, and its content in soil varies across different regions. In many countries, including China, selenium-containing soil is poor, and people are prone to selenium deficiency [9,10]. Selenium can exist in nature as inorganic forms (e.g., selenate and selenite) and organic forms (e.g., selenocysteine [Sec]) [11]. Selenium is highly absorbed and distributed throughout the body; in particular, organic selenium is more stable and bioavailable than inorganic selenium [11]. Sec is the main form of selenium in cells, and Sec-containing protein, namely selenoprotein (e.g., selenoprotein P, glutathione peroxidases [GPxs], and thioredoxin reductase [TrxRs]), is mainly responsible for the biological role of selenium in the human body [11,12,13,14]. Selenium is incorporated into selenoproteins, which have broad pleiotropic functions, such as antioxidant and anti-inflammatory properties [15]. Oxidative stress, which is an imbalance between antioxidant defense and prooxidant substances (e.g., reactive oxygen species [ROS] and reactive nitrogen species [RNS]), causes oxidative damage through various mechanisms, including lipid peroxidative damage, DNA damage, and protein oxidation [16,17]. Lately, increasing evidence has shown that the progression of insulin resistance, the pathogenesis of T2DM, and its microvascular ailments and macrovascular complications are significantly regulated by oxidative stress [18,19]. Higher oxidative stress is directly related to the emergence of CMDs [20,21]. Selenium is a well-known antioxidant; in particular, the selenoproteins GPxs and TrxRs are involved in antioxidant defenses and protection against oxidative damage [22]. Supplementing selenium could significantly reduce ROS, increase superoxide dismutase and GPxs activity, and reduce inflammatory cytokine content [23]. A meta-analysis of 13 trials found that selenium supplementation alleviated oxidative stress by raising the total antioxidant capacity and GPxs levels and lowering serum malonaldehyde [17]. It has been proposed that selenium is a hormetic chemical, a substance with a biphasic dose-response that is poisonous at high levels but beneficial at low concentrations [22]. Supra-nutritional levels of selenium produce ROS, which then disturb the redox states of cells [24], increase oxidative stress, and damage tissues and organs [25]. Therefore, maintaining an optimal selenium status is crucial to maintaining redox equilibrium.

Numerous studies have highlighted the significance of selenium and selenoproteins in the prevention and treatment of chronic metabolic diseases such as MS, T2DM, and CVD [26,27,28]. Huang et al. [29] reported that low selenium levels were related to an elevated risk of metabolic disorders, poor prognosis, and mortality. Interestingly, Kamali et al. [30] observed that selenium supplementation significantly improved glucose metabolism by decreasing fasting plasma glucose (FPG), insulin, and homeostatic model assessment of insulin resistance (HOMA-IR), and also increased high-density lipoprotein-cholesterol (HDL-C) levels, but did not affect other lipid profiles. However, selenium status has been reported to be positively associated with markers of insulin resistance and lipid profiles by Cardoso et al. [31] and Ju et al. [32]. On the other hand, a previous meta-analysis reported that selenium supplementation significantly alleviated oxidative stress and inflammation, but did not improve the blood lipid status [33]. To the best of our knowledge, the exact role of selenium in glycolipid metabolism in patients with CMDs remains undetermined. Therefore, to address these issues, we analyzed the impacts of selenium supplementation on glucose and lipid metabolism in CMDs, with the aim of verifying whether selenium supplementation could be a complementary treatment strategy for CMDs.

## 2. Methods

This meta-analysis strictly followed the Preferred Reporting Items for Systematic Reviews and Meta-Analyses (PRISMA) guidelines [34]. The study protocol has been registered and published in PROSPERO with ID: CRD42022353393.

### 2.1. Search Strategy

Searches of the literature for this meta-analysis were conducted using PubMed, Cochrane Library, Embase, Scopus, and Web of Science databases up to 31 July 2022. The key search terms for searching the databases included the following: selenium, selenite, selenate, trace element, cardiometabolic disease, diabetes mellitus, T2DM, coronary heart disease, heart failure, hypertension, hyperlipidemia, metabolic syndrome, stroke, obesity, randomized controlled trial, RCT, random*. In some cases, we may have added or changed the retrieved keywords depending on the characteristics of the databases (Appendix A). Moreover, we manually checked the reference lists of the eligible articles to identify extra pertinent research. Two reviewers (J.O. and Y.C.) conducted the literature search independently, and any discrepancies were resolved by consensus.

### 2.2. Study Selection

Two authors (J.O. and Y.C.) individually filtered all eligible studies using strict inclusion and exclusion criteria. Any differences in opinion were settled through consensus or discussion with Drs. Bai and Wang. The reasons for the exclusion of studies in each phase were recorded. Eligible studies were required to meet the following inclusion criteria according to PICOS: (1) Types of population (P): patients with diseases related to CMDs, such as diabetes mellitus, coronary heart disease, heart failure, hypertension, stroke, metabolic syndrome, and obesity. (2) Types of interventions (I): the experimental group received selenium supplementation but the control group did not. Selenium supplementation in all forms, including inorganic, organic, synthetic, and selenium-enriched yeast, was considered. The treatment dose and period were not limited. (3) Types of comparison (C): the control group received placebo or conventional treatment. (4) Types of outcomes (O): primary outcomes: insulin levels, HOMA-IR, FPG, and glycosylated hemoglobin A1C (HbA1c); secondary outcomes: lipid profiles, including total cholesterol (TC), triglycerides (TG), low-density lipoprotein cholesterol (LDL-C), very low-density lipoprotein cholesterol (VLDL-C), and HDL-C. (5) Types of study design (S): randomized controlled trial (RCT) only. The exclusion criteria were as follows: (1) Repeat published studies; (2) conference abstracts; and (3) in vitro and animal studies.

### 2.3. Data Collection Process

Two reviewers (J.O. and Y.C.) independently collected the following data from the included studies using standardized forms: (1) the characteristics of selected articles, such as author(s), journal of publication, publication year, study design, study location, registration, or not, the number of participants, interventions, treatment period; (2) characteristics of participants, such as disease type, mean age, gender; and (3) clinical outcomes.

### 2.4. Risk of Bias Assessment

We assessed the risk of bias of the included studies according to the Cochrane Collaboration Risk of Bias tool. The assessed domains included the following: methods of random sequence generation, allocation concealment, blinding of participants and personnel, blinding of outcome assessment, incomplete outcome data, selective reporting, and other bias [35]. Each study was classified as low, unclear, or high risk based on these domains.

### 2.5. Data Synthesis and Statistical Analysis

The effect of selenium supplementation on relevant outcomes was assessed as the changes (mean ± standard deviation [SD]) before and after treatment in the experimental and the control groups. If the mean values of the changes before and after treatment were unreported, they were calculated by subtracting the mean at the baseline from the mean at the end of the follow-up. When the SDs of the changes before and after treatment were not reported, they were computed according to the number of patients, standard errors, 95% confidence interval (CI), interquartile ranges, or *p*-values. If the missing SDs were still unavailable, they were calculated using the correlation formula, and the correlation coefficient was cautiously assumed to be 0.5 [36,37]. For studies with multiple intervention groups, we combined relevant groups into a single treatment group. All related calculation formulas were referred to the Cochrane Handbook for Systematic Reviews of Intervention [38].

Data were evaluated using Review Manager version 5.3 and STATA version 17.0 for a more comprehensive assessment of outcomes. The heterogeneity between studies was assessed using Cochrane’s Q test and was quantified by the *I*^2^ test. Heterogeneity was rated as low, moderate, or high when the value of *I*^2^ was <50%, 50–75%, or >75%, respectively [39]. When the heterogeneity was low (*I*^2^ < 50%), data were pooled by applying the fixed-effects model; otherwise, the random-effects model was applied [40]. Effect sizes are presented as the standardized mean difference (SMD) with 95% CI. If sufficient studies (≥10) were included, funnel plots and Egger’s test were applied to determine whether there was publication bias. A *p*-value of <0.05 was considered statistically significant.

### 2.6. Analysis of Subgroups or Subsets

In cases where significant heterogeneity was noted among studies, sensitivity analysis or subgroup analyses were performed to identify its possible sources. Sensitivity analysis was performed by removing each study sequentially to evaluate the influence of each study on the overall effect size. Subgroup analysis was conducted according to the type of disease of the participants.

## 3. Results

### 3.1. Literature Selection

We retrieved 4688 studies from five electronic databases. Two studies were manually retrieved. Then, 2530 studies were retained after excluding 2160 duplicates, and a further 2503 studies were eliminated after reading the title and abstract, leaving 27 studies that met the screening criteria for full-text evaluation. Finally, 10 RCTs [30,41,42,43,44,45,46,47,48,49] were included in this meta-analysis. The screening process is depicted in Figure 1.

### 3.2. Study Characteristics

All 10 studies included in this meta-analysis were randomized, double-blind, placebo-controlled trials, with 526 participants, including 272 in the selenium group (experimental group) and 254 in the control group. The treatment period ranged from 4 to 24 weeks. All 10 studies were conducted in Iran. Except for Faghihi 2014 [48], the remaining nine studies have completed clinical trial registration. Faghihi 2014 [48] reported participants’ selenium concentration as deficient state at baseline, and the remaining studies did not report participants’ selenium status. In the included studies, the forms of selenium supplementation were mainly selenium yeast and sodium selenite, but three studies did not mention the form of selenium supplementation. Five studies [37,41,43,45,48] recruited patients with diabetes mellitus or complications of diabetes mellitus (e.g., diabetic nephropathy), three studies [30,42,44] recruited patients with cardiovascular disease, one study [46] recruited patients with diabetes mellitus combined with coronary heart disease, and one study [49] recruited obese patients (Table 1).

### 3.3. Risk of Bias Assessments

All except two studies [46,49] were rated as having a low risk of selection bias for adopting appropriate random sequence generation and allocation concealment methods. Farrokhian 2016 [46] and Alizadeh 2012 [49] were assessed as having an unclear risk of selection bias because they reported random sequence generation methods, but did not report allocation concealment methods. All of the 10 studies were rated as carrying a low risk of performance bias and concealment bias due to complete reporting of blinding implementation. Eight studies [30,41,42,44,45,46,47,49] were rated as having a low risk of attrition bias, but Najib 2020 [43] and Faghihi 2014 [48] were rated as high risk due to unbalanced and unexplained loss at follow-up. Farrokhian 2016 [46] and Alizadeh 2012 [49] were rated as high risk of reporting bias because the primary outcomes were changed after the protocol registration. Due to the lack of a registered protocol [48] or the inability to report several secondary outcomes [41,42,43,45], five studies were rated as unknown risks of report bias. Farrokhian 2016 [45] was rated a high risk of other bias because of the inconsistency in the types of hypoglycemic drugs taken between the selenium and control groups, which may have affected the effect evaluation (Figure 2 and Figure 3).

### 3.4. Meta-Analysis

#### 3.4.1. Primary Outcomes

All 10 studies with 526 participants evaluated the effects of serum insulin levels and HOMA-IR. The heterogeneity of serum insulin levels and HOMA-IR were moderate (*I*^2^ = 68%, *I*^2^ = 75%). Pooled results obtained by employing a random-effects model demonstrated that selenium supplementation remarkably lowered serum insulin levels (SMD: −0.53, 95% CI [−0.84, −0.21], *p* = 0.001) and decreased HOMA-IR (SMD: −0.50, 95% CI [−0.86, −0.14], *p* = 0.006) (Figure 4 and Figure 5). To resolve heterogeneity, sensitivity analysis was conducted by excluding the studies one by one. The pooled results were broadly consistent with the above analysis (Appendix A), and the heterogeneity was largely affected by Faghihi 2014 [48], which was excluded. Nine studies [30,41,42,43,44,45,46,47,49] remained after the exclusion, with no heterogeneity in the pooled results (all *I*^2^ = 0%), indicating that Faghihi 2014 [48] was a major factor in the source of heterogeneity of insulin levels and HOMA-IR. This may be due to a baseline difference in the hypoglycemic drugs taken between the selenium and control groups in Faghihi’s study. An analysis was then conducted with the fixed-effects model, and the result confirmed the previous observation that supplementing with selenium was associated with lower serum insulin levels (SMD: −0.67, 95% CI [−0.86, −0.48], *p* < 0.0001) and HOMA-IR (SMD: −0.67, 95% CI [−0.86, −0.48], *p* < 0.0001) (Figure 6 and Figure 7).

The effect of selenium supplementation on FPG was assessed in 492 participants through nine studies [30,41,42,43,44,45,46,47,48]. The heterogeneity between studies was high (*I*^2^ = 91%). Pooled analysis from the random-effects model indicated that the selenium group and the control group had similar effects on FPG (SMD: 0.06, 95% CI [−0.56, 0.68], *p* = 0.86) (Appendix A). To resolve heterogeneity, sensitivity analysis was conducted by excluding the studies one by one. The results showed that although the pooled results were stable, the heterogeneity was not resolved (Appendix A). Then, subgroup analysis was conducted based on the underlying diseases of the participants. As shown in Figure 8, the FPG levels in patients with cardiovascular disease were significantly lower in the selenium group than in the control group (SMD: −0.42, 95% CI: [−0.77, −0.07], *p* = 0.02), with no heterogeneity (*I*^2^ = 0%). However, there was no statistical difference in terms of FPG between the selenium and control groups in the other two subgroups (Figure 8).

Only two studies [43,48], including 114 participants, assessed the effect of selenium supplementation on HbA1c. The heterogeneity between studies was high (*I*^2^ = 85%). Thus, we did not perform a meta-analysis of HbA1c. Both studies reported reductions in HbA1c after treatment in both the selenium and control groups, in which Najib 2020 [43] reported a more significant reduction in HbA1c in the selenium group compared to the control group.

#### 3.4.2. Secondary Outcomes

Nine studies [30,41,42,44,45,46,47,48,49] with 492 patients evaluated the effects of TC, TG, and LDL-C, and five studies [30,41,44,45,46] with 259 patients evaluated the effects of VLDL-C. The heterogeneity of TC, TG, and VLDL-C was insignificant (all *I*^2^ = 0). Unfortunately, the pooled results from the fixed-effects model demonstrated that selenium supplements did not significantly lower TC, TG, and VLDL-C in patients with CMDs (SMD: −0.07, 95% CI [−0.25, 0.12], *p* = 0.48, SMD: −0.12, 95% CI [−0.30, 0.06], *p* = 0.20, and SMD: −0.08, 95% CI [−0.33, 0.16], *p* = 0.51, respectively) (Figure 9, Figure 10 and Figure 11). The heterogeneity of LDL-C was high (*I*^2^ = 79%). Pooled results from the random-effects model demonstrated no significant difference in LDL-C between the two groups (SMD: −0.35, 95% CI [−0.76, 0.06], *p* = 0.10) (Figure 12).

A total of nine studies [30,41,42,44,45,46,47,48,49], including 492 patients, evaluated the effects of HDL-C. The pooled results from the random-effects model indicated that selenium supplementation remarkably increased HDL-C levels (SMD: 0.97, 95% CI [0.26, 1.68], *p* = 0.007), with high heterogeneity across studies (*I*^2^ = 92%) (Appendix A). To resolve heterogeneity, a sensitivity analysis was conducted by excluding each study separately. The results showed that the pooled results were broadly consistent with the above analysis (Appendix A), and the heterogeneity was largely affected by Ghazi 2021 [42]. Eight studies [30,41,44,45,46,47,48,49] remained after excluding Ghazi 2021 [42], and pooled results showed that the heterogeneity between studies was decreased (*I*^2^ = 58%) (Appendix A). The participants of Ghazi 2021 [42] were patients with atherosclerosis, and dyslipidemia is closely related to atherosclerosis. Therefore, considering that the source of heterogeneity may be related to the participants’ underlying disease, the included studies were divided into subgroups based on the participants’ disease type. As shown in Figure 13, the HDL-C levels were significantly increased in the diabetes mellitus subgroup and cardiovascular disease subgroup (SMD: 0.50, 95% CI [0.10, 0.91], *p* = 0.02 and SMD: 4.22, 95% CI [1.06, 7.37], *p* = 0.009), with significant heterogeneity among studies (*I*^2^ = 59% and *I*^2^ = 97%). However, there was no statistical difference between the selenium and control groups in the other two subgroups (Figure 13).

### 3.5. Publication Bias

Funnel plots of insulin levels and HOMA-IR were drawn with Review Manager version 5.3, and Egger’s test was conducted to quantify the publication bias with Stata version 17.0. The two funnel plots were substantially symmetrical (Figure 14). The results of Egger’s test for insulin levels and HOMA-IR were *p* = 0.678 and *p* = 0.908, respectively, indicating that there was no publication bias.

## 4. Discussion

Over the last decade, increasing attention has been paid to the selenium status in patients with various cardiometabolic diseases and the association between selenium status and mortality risk of various cardiometabolic diseases. Several meta-analyses have indicated that individuals with cardiometabolic diseases tend to have lower selenium levels than healthy individuals [50,51,52], and that selenium levels are negatively associated with mortality among patients with MS, T2DM, and CVD [53,54,55]. Alterations in the metabolism of glucose and lipids characterize metabolic disorders [56]. Insulin has regulatory effects on glucose, which are mainly classified into two aspects: promoting glucose absorption into liver cells, muscle cells, and adipose tissue, and inhibiting glycogenolysis and gluconeogenesis in the liver [57]. Insulin resistance is a common pathophysiological status in which individuals have decreased insulin sensitivity and impaired glucose regulation by insulin, resulting in glucose intolerance [58]. It is well-established that insulin resistance underpins many chronic metabolic diseases [59], and cardiometabolic risks such as obesity and dyslipidemia can exacerbate insulin resistance and impel the progression of CMDs [1]. Studies have shown that the micronutrient selenium can regulate the human body’s insulin sensitivity, and selenium in the form of selenate is known to act as an insulin mimetic with a role in maintaining blood sugar homeostasis [60,61]. Expression of selenoprotein *P* plays a crucial role in pancreatic β-cell function by preventing ferroptosis and thus maintaining glutathione peroxidase 4 (Gpx4) and cell viability, and also by inhibiting stress-induced degradation of nascent granules, a novel regulatory pathway for insulin [62], thus maintaining cellular proinsulin levels [63]. Furthermore, a meta-analysis performed by Tabrizi et al. [64] supported the positive effect of selenium supplements on lipid profiles. Of note, that selenium supplementation makes the most sense when there is deficit of selenium [14,65].

In this meta-analysis, we examined the impact of selenium supplementation on the markers of insulin resistance, glucose, as well as blood lipid profiles in patients with CMDs. The level of glucose in the blood is one of the most significant homeostatic indicators and is strictly regulated [66]. The pathways and regulation of glucose metabolism include glycolysis/glycogenolysis, gluconeogenesis, pentose phosphate pathway (PPP), insulin signaling pathway, and many others [67], some of which can be regulated by insulin [57]. HOMA-IR is often used in investigating and quantifying insulin resistance because of the simplicity of the underlying mathematical model (HOMA-IR = fasting glucose [mg] × fasting insulin [mu/L]/22.5) [68]. Furthermore, HbA1c is an essential marker of long-term glycemic control that reflects a cumulative glycemic level over the past two to three months [69]. Therefore, insulin levels, HOMA-IR, FPG and HbA1c were used to evaluate the selenium supplementation on glycemic control. In this study, comprehensive pooled results from 10 RCTs involving 526 patients supported the favorable effects of selenium supplementation in decreasing serum insulin levels and HOMA-IR. Moreover, selenium supplementation may increase HDL-C levels, but the effectiveness of selenium supplementation on FPG, TC, TG, LDL-C, and VLDL-C levels was unclear. The current results suggest that selenium supplementation may be an effective treatment for reducing insulin resistance.

The findings reveal that selenium supplementation could reduce insulin levels and HOMA-IR in patients with CMDs, but the effect on FPG was ambiguous. This result is similar to that of the meta-analysis conducted by Strozyk et al. in 2017, which included four RCTs [70]. Their study focused on assessing the efficacy of selenium supplementation in T2DM, and the results supported that selenium supplementation significantly improved insulin resistance. However, our study also focused on other cardiometabolic diseases, such as cardiovascular disease, and collected RCTs that have been updated recently. Therefore, 10 RCTs were included in this meta-analysis. The pooled results derived from the included studies are in line with those of previous animal studies [71,72,73], in that supplemental selenium therapy had a significantly protective anti-diabetic effect. Selenium nanoparticles (selenium-NPs) are a new supplemental form of selenium that is readily absorbed and has low toxicity. According to Abdulmalek et al. [74], treating diabetic rats with selenium-NPs (0.1 and 0.4 mg/kg) and/or metformin (100 mg/kg) separately or together, led to a remarkable decrease in FBG and insulin levels, suggesting that selenium and its derivatives play a significant role in the maintenance of glucose homeostasis. There is a lack of positive evidence regarding the effect of selenium supplementation on HbA1c. A cross-sectional study reported that obese participants with lower selenium intakes exhibited higher concentrations of HbA1c [75]. In this regard, the effect of selenium supplementation on long-term glycemic control deserves further attention.

The pooled results of this study also demonstrated that selenium supplementation increased HDL-C levels, but had little effect on other blood lipids. In addition, in the subgroup of disease types, we found that selenium supplementation increased HDL-C levels more significantly in participants with cardiovascular disease, followed by those with diabetes mellitus. HDL-C is essential for reverse cholesterol transport and has anti-inflammatory, anti-atherogenic, and anti-thrombotic effects [76]. The interaction of these properties contributes to the cardioprotective properties of HDL-C. Thus, appropriate selenium supplementation may contribute to the improvement of CMDs, particularly cardiovascular disease.

Some statistical heterogeneity was discovered in the pooled analysis of insulin levels, HOMA-IR, FPG, and HDL-C. The results were similar before and after sensitivity analysis, suggesting that the results were stable and reliable. In terms of insulin levels and HOMA-IR outcomes, we considered that Faghihi’s study [48] administered inconsistent types or combinations of antidiabetic drugs at baseline, which might be an essential source of heterogeneity in the pooled analysis. In Faghihi’s study [48], approximately 85.2% of participants in the control group received combined antidiabetic medication, compared to only 66.6% of participants in the selenium group. Additionally, subgroup analysis suggested that the inconsistency of participants’ underlying diseases is a major source of heterogeneity in terms of FPG and HDL-C. In the future, more evaluable studies should be included in the analysis to better systematically evaluate the effectiveness of selenium supplementation in different types of CMDs.

This meta-analysis has several strengths. First, two reviewers independently used a comprehensive search strategy to identify all trials evaluating the effect of selenium supplementation in patients with CMDs and used standardized templates to extract data from included trials to guarantee data accuracy and reduce the impact of study design faults. Second, most included RCTs were high-quality with appropriate randomization, allocation concealment, and double-blinding methods. Third, no publication bias was found in this meta-analysis. Furthermore, we performed a thorough sensitivity analysis to examine the stability of our results. However, there are several limitations that should be considered. First, as the control group was placebo instead of different doses of selenium, the optimal dose of selenium supplementation cannot be accurately determined in this study. Second, the number of participants in the included RCTs was relatively small, with none having more than 100. Third, as only two studies evaluated HbA1c and there was high heterogeneity in the pooled analysis results, we could only perform a systematic review, which may affect the reliability and comprehensiveness of the evaluation of the efficacy of selenium supplementation on the HbA1c control. Fourth, although trials of the effects of selenium supplementation on CMDs have been conducted in countries other than Iran, trials in which other nutritional supplements were supplemented in addition to selenium in the intervention [77], and trials in which markers related to glucolipid metabolism were not reported [78] were excluded from this meta-analysis, and the final relevant included studies were all conducted in Iran, which may limit the generalizability. According to this, the recommendations of potential effects of selenium supplementation conclusions should be drawn with caution. Finally, we suggest that future clinical trials should pay more attention to the different doses of selenium supplements and the baseline level of serum selenium of CMDs patients, in order to further illuminate the therapeutic effects of selenium supplementation on CMDs.

## 5. Conclusions

This meta-analysis demonstrated that selenium supplementation may reduce the levels of serum insulin and HOMA-IR, and increase serum HDL-C levels, suggesting that selenium supplementation may be beneficial for reducing insulin resistance in patients with CMDs. This finding may provide support to prospective studies looking into selenium supplementation to manage cardiometabolic risk factors. However, in the case of selenium excess, the efficacy of selenium supplementation on glucolipid metabolism needs further evaluation.

## Figures and Tables

**Figure 1 nutrients-14-04933-f001:**
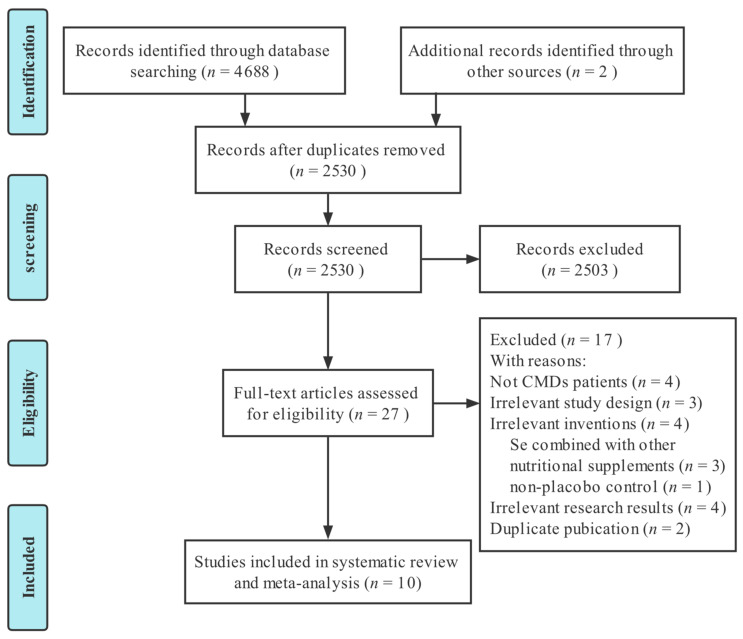
PRISMA flow chart for selection and screening of the studies.

**Figure 2 nutrients-14-04933-f002:**
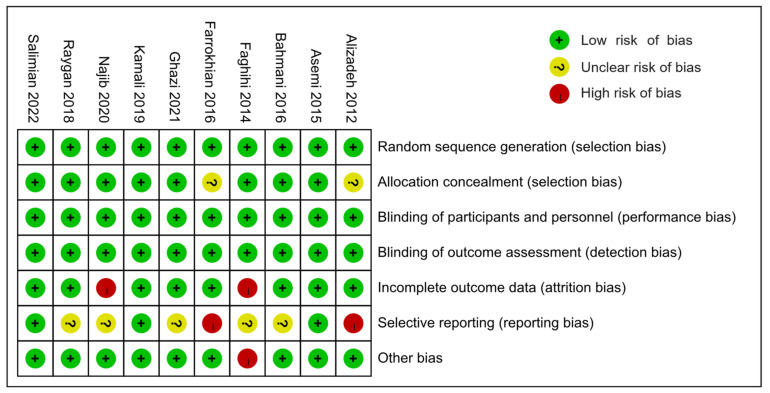
Summary of the risks of bias.

**Figure 3 nutrients-14-04933-f003:**
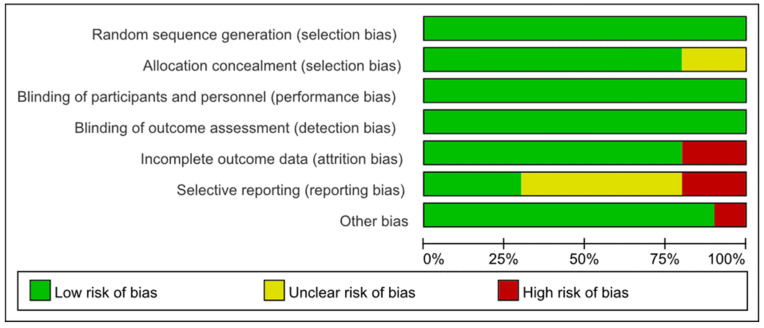
Risks of bias graph expressed as percentages.

**Figure 4 nutrients-14-04933-f004:**
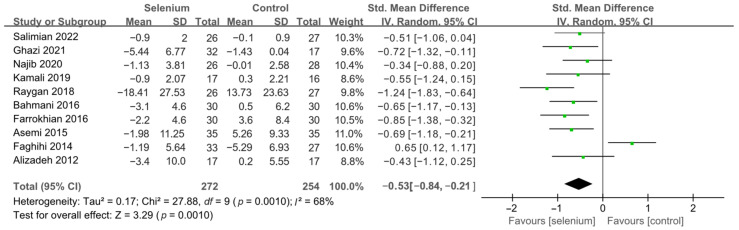
Forest plot of insulin levels [30,41,42,43,44,45,46,47,48,49].

**Figure 5 nutrients-14-04933-f005:**
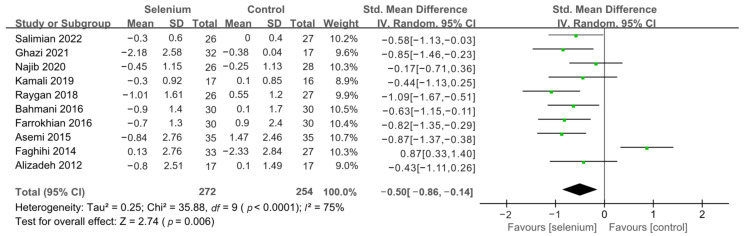
Forest plot of HOMA-IR [30,41,42,43,44,45,46,47,48,49].

**Figure 6 nutrients-14-04933-f006:**
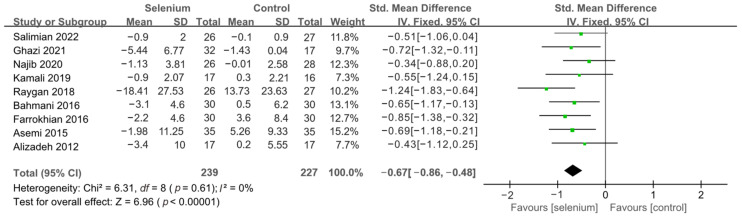
Forest plot of insulin levels after excluding Faghihi 2014 [30,41,42,43,44,45,46,47,49].

**Figure 7 nutrients-14-04933-f007:**
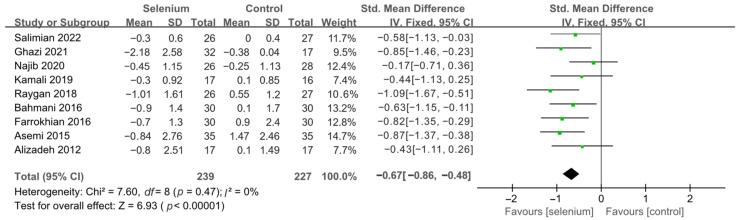
Forest plot of HOMA-IR after excluding Faghihi 2014 [30,41,42,43,44,45,46,47,49].

**Figure 8 nutrients-14-04933-f008:**
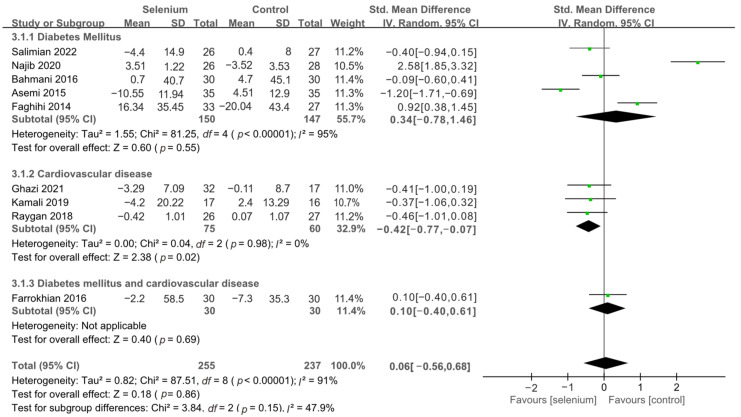
Subgroup analysis of FPG [30,41,42,43,44,45,46,47,48].

**Figure 9 nutrients-14-04933-f009:**
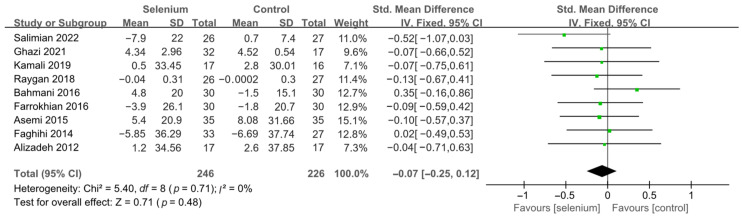
Forest plot of TC [30,41,42,44,45,46,47,48,49].

**Figure 10 nutrients-14-04933-f010:**
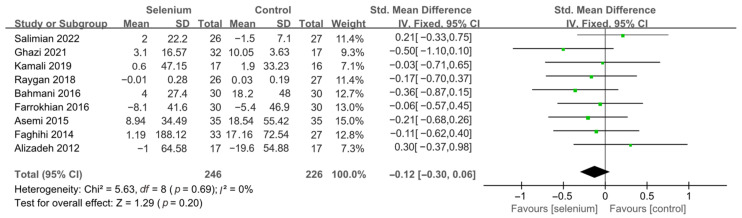
Forest plot of TG [30,41,42,44,45,46,47,48,49].

**Figure 11 nutrients-14-04933-f011:**
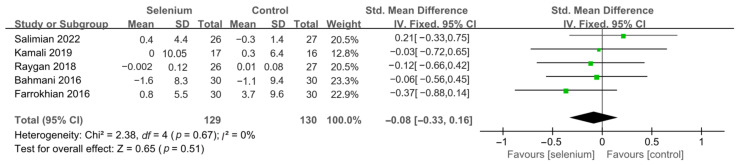
Forest plot of VLDL-C [30,41,44,45,46].

**Figure 12 nutrients-14-04933-f012:**
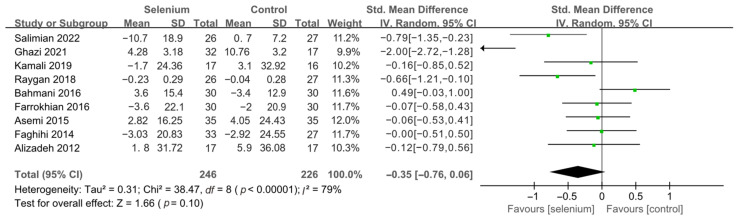
Forest plot of LDL-C [30,41,42,44,45,46,47,48,49].

**Figure 13 nutrients-14-04933-f013:**
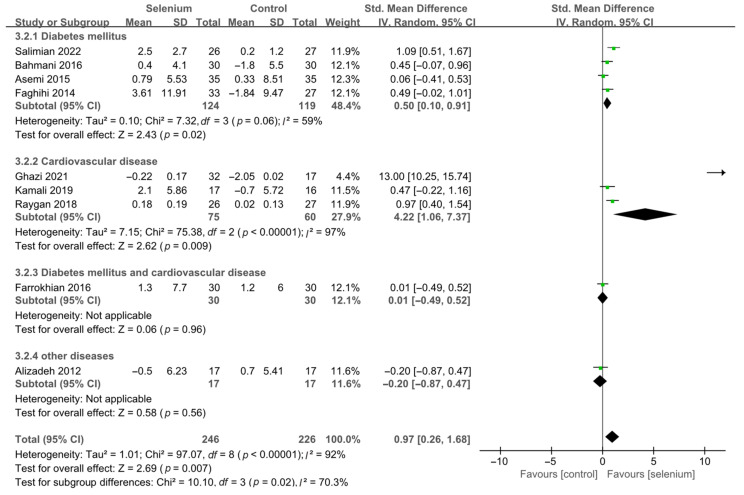
Subgroup analysis of HDL-C [30,41,42,44,45,46,47,48,49].

**Figure 14 nutrients-14-04933-f014:**
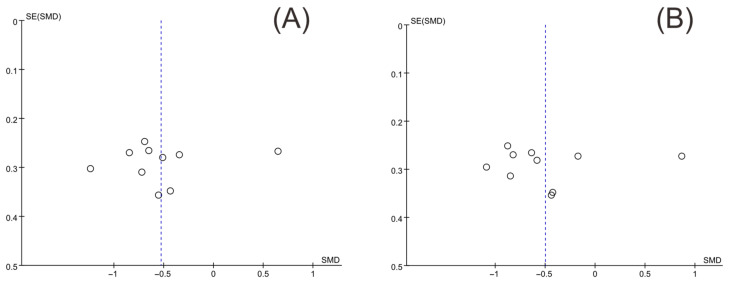
Funnel plot of insulin levels and HOMA-IR. (**A**) Insulin levels, (**B**) HOMA-IR.

**Table 1 nutrients-14-04933-t001:** Characteristics of the included studies.

Study	Study Location	Study Design	Groups	No. of Participants	Mean Age	Gender (M/F)	Intervention	Treatment Period	Type of Disease
Salimian 2022 [41]	Iran	Single-center double-blinded RCT	I	26	57.6 ± 11.5	NR	selenized yeast 200 μg/day	24 weeks	Diabetic nephropathy
			C	27	61.5 ± 9.8	NR	placebo		
Ghazi 2021 [42]	Iran	Single-center double-blinded RCT	I_1_	16	59.06 ± 8.55	12/4	selenium-enriched yeast 200 μg/day	8 weeks	Atherosclerosis
			I_2_	16	58.62 ± 9.68	16/0	sodium selenite 200 μg/day		
			C	17	53.58 ± 13.75	15/2	Placebo		
Najib 2020 [43]	Iran	Multi-center double-blinded RCT	I	26	29.19 ± 6.16	0/26	selenium supplements 100 μg/day	12 weeks	Gestational diabetes mellitus
			C	28	31.0 ± 4.43	0/28	Placebo		
Kamali 2019 [30]	Iran	Single-center double-blinded RCT	I	17	62.6 ± 11.6	NR	selenium yeast 200 μg/day	4 weeks	Coronary heart disease
			C	16	61.2 ± 4.6	NR	Placebo		
Raygan 2018 [44]	Iran	Single-center double-blinded RCT	I	26	70.7 ± 10.3	8/18	selenium yeast 200 μg/day	12 weeks	Congestive heart failure
			C	27	68.5 ± 7.7	8/19	Placebo		
Bahmani 2016 [45]	Iran	Single-center double-blinded RCT	I	30	63.1 ± 12.6	15/15	selenium supplements 200 μg/day	12 weeks	Diabetic nephropathy
			C	30	61.4 ± 9.3	15/15	Placebo		
Farrokhian 2016 [46]	Iran	Single-center double-blinded RCT	I	30	NR	10/20	selenium yeast 200μg/day	8 weeks	Type 2 diabetes mellitus and coronary heart disease
			C	30	NR	10/20	Placebo		
Asemi 2015 [47]	Iran	Single-center double-blinded RCT	I	35	27.6 ± 5.3	0/35	selenium supplements 200 μg/day	6 weeks	Gestational diabetes mellitus
			C	35	29.6 ± 3.6	0/35	Placebo		
Faghihi 2014 [48]	Iran	Single-center double-blinded RCT	I	33	53.54 ± 7.52	16/17	sodium selenite 200 μg/day	3 months	Type 2 diabetes mellitus
			C	27	55.76 ± 7.77	18/9	Placebo		
Alizadeh 2012 [49]	Iran	Single-center double-blinded RCT	I	17	36.6 ± 8.6	0/17	selenium-enriched yeast 200 μg/day	6 weeks	Obesity
			C	17	36.7 ± 8.3	0/17	Placebo		

M: Male, F: Female, I: Intervention, C: Control, NR: Not reported, RCT: Randomized controlled trial.

## Data Availability

Not applicable.

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
