# Peer review of "Potential Benefits of Selenium Supplementation in Reducing Insulin Resistance in Patients with Cardiometabolic Diseases: A Systematic Review and Meta-Analysis"

_nutrients, 2022, doi:10.3390/nu14224933_

Round 1

Reviewer 1 Report

The meta-analysis has been conducted according to the existing standards. Surprisingly, only Iranian studies could be retrieved. Is there an explanation for this? Is it conceivable that this type of research is performed properly only in Iran?

All abbreviations should be explained at first appearance (SMD in the  abstract).

Author Response

Dear reviewer,

We highly appreciate your valuable comments concerning our manuscript entitled “Potential benefits of selenium supplementation in improving insulin resistance in patients with cardiometabolic diseases: a systematic review and meta-analysis” (Manuscript ID: nutrients-2001701). We have revised the manuscript carefully in light of helpful comments from you and the editor. All revisions in the manuscript using revision mode, and details point-by-point replies are attached.

Best regards,

Yours sincerely,

Anlu Wang, and Ruina Bai

Xiyuan Hospital, China Academy of Chinese Medical Sciences

2022.11.11

Reviewer 2 Report

The meta-analysis, authored by Ouyang J. et al. aimed to identify the effects of selenium supplementation on insulin resistance, glucose homeostasis, and lipid profiles in patients with cardiometabolic diseases (CMD).

Overall methodologically review and meta-analysis is performed quite well and data is presented clearly, however, some major issues are related with the general concept of the manuscript. Some of them I would like to summarize below:

1. Selenium is neither "good" nor "bad" with regard of health effects. When there is deficit of Se due to poor nutrition or endemic lack of this mineral in soil then supplementation would make most sense. While excessive amounts of Se are toxic. So enerally speaking supplementation with potential therapeutic effect makes only sense when there is documented deficit or at least deficit is expected.

2. Insulin resistance/sensitivity is very complex phenomenon and effect of Se should make sense mechanistically. Glucose homeostasis is even more sophisticated. This requires clear hypothesis why these and not the other parameters were selected for meta-analysis. There is plenty of literature that may help, for example PMID: 32033390.

3. In terms of clinically meaningful and validated benefit that is usually required or expected to be estimated for clinical trials, insulin resistance estimated as HOMA-IR may not be perfect. HbA1c may be best among the parameters of glucose control, but it was not evaluated. FBG is not perfect as well.

4. Conclusions must be rewritten to reflect limited evidence of "clinical benefit", but effects on fasting insulin levels and HOMA-IR may support future prospective studies.

5. Limitations of the study must be more clearly indicated in discussion section.

Minor: title:  improving insulin resistence is not perfect, I suggest "improving insulin sensitivity" or "reducing insulin resitance".

Author Response

Dear Reviewer,

We highly appreciate your constructive comments of our manuscript entitled “Potential benefits of selenium supplementation in improving insulin resistance in patients with cardiometabolic diseases: a systematic review and meta-analysis” (Manuscript ID: nutrients-2001701). We have revised the manuscript carefully in light of helpful comments from you and the editor. All revisions in the manuscript using revision mode, and details point-by-point replies are attached. 

Best regards,

Yours sincerely,

Anlu Wang, and Ruina Bai

Xiyuan Hospital, China Academy of Chinese Medical Sciences

2022.11.11

Round 2

Reviewer 2 Report

It seems that article type "Pespective" is not appropriate in this case. It should be Review/Meta Analysis" or something similar, as it reflects the nature of this work.